# TerDiT: Ternary Diffusion Models with Transformers

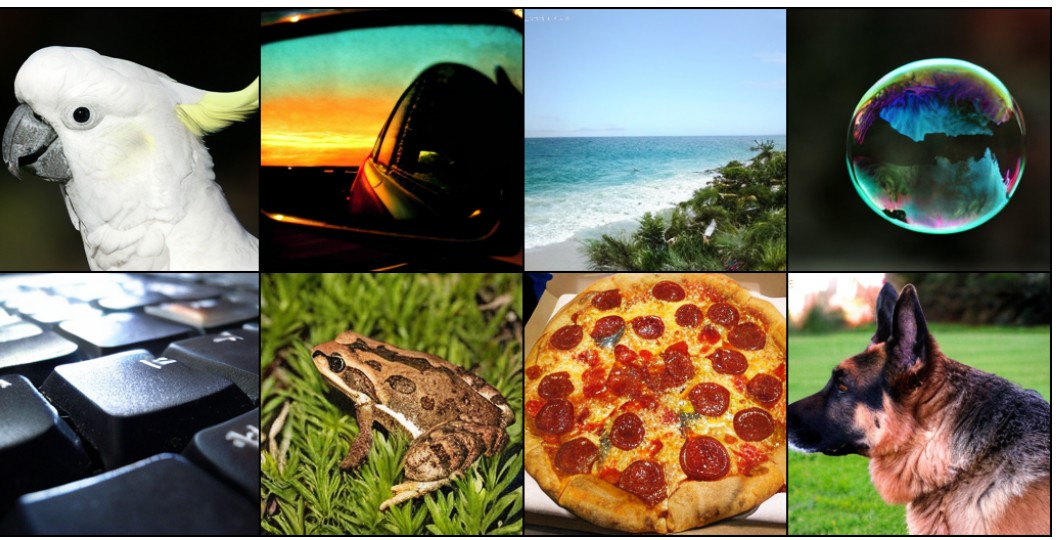

Figure 1: Sample images (256×256) generated by the ternary DiT model with 4.2B parameters (using 2GB of GPU memory) are shown. For comparison, images generated by full-precision diffusion transformer models—DiT-XL/2 with 675M parameters (using 3GB of GPU memory) and Large-DiT-4.2B with 4.2B parameters (using 17GB of GPU memory)—are provided in Fig. 4.

## Abstract

Recent developments in large-scale pre-trained text-to-image diffusion models have significantly improved the generation of high-fidelity images, particularly with the emergence of diffusion transformer models (DiTs). Among diffusion models, diffusion transformers have demonstrated superior image generation capabilities, boosting lower FID scores and higher scalability. However, deploying large-scale DiT models can be expensive due to their excessive parameter numbers. Although existing research has explored efficient deployment techniques for diffusion models such as model quantization, there is still little work concerning DiT-based models. To tackle this research gap, in this paper, we propose **TerDiT**, a quantization-aware training (QAT) and efficient deployment scheme for ternary diffusion transformer models. We focus on the ternarization of DiT networks, with model sizes ranging from 600M to 4.2B, and image resolution from 256×256 to 512×512. Our work contributes to the exploration of efficient deployment of large-scale DiT models, demonstrating the feasibility of training extremely low-bit DiT models from scratch while maintaining competitive image generation capacities compared to full-precision models. Code has been uploaded in the supplemental materials.

## 1 Introduction

The advancements in large-scale pre-trained text-to-image diffusion models (Ho et al., 2020; 2022a; Ramesh et al., 2022; Rombach et al., 2022; Saharia et al., 2022) have led to the successful generation of images characterized by both complexity and high fidelity to the input conditions. Notably, the

emergence of diffusion transformer models (DiTs) (Peebles & Xie, 2023) represents a significant stride in this research direction. Compared with other diffusion models, diffusion transformers have demonstrated the capability to achieve lower FID scores but with higher computation GFLOPS (Peebles & Xie, 2023). Recent research highlights the remarkable image and video generation capabilities of diffusion transformer architectures, as demonstrated in methods like Stable Diffusion 3 (Esser et al., 2024) and Sora (Brooks et al., 2024).

Given the impressive performance of diffusion transformer models, researchers are now training larger and larger DiTs (Liu et al., 2024). For instance, Stable Diffusion 3 trained DiT models ranging from 800 million to 8 billion parameters. Additionally, there is speculation among researchers that Sora might boast around 3 billion parameters. Given the enormous parameter numbers, deploying these DiT models will be costly, especially on certain end devices (e.g., mobile phones).

To deal with the deployment dilemma, there were recent works on the efficient deployment of diffusion models (Li et al., 2023; 2024b; He et al., 2024; Wang et al., 2024), most of which focus on model quantization. However, as far as we are concerned, there are still two main shortcomings in current research. **Firstly**, the exploration of quantization methods for transformer-based diffusion models remains quite limited (Chen et al., 2024; Deng et al., 2024) compared to quantizing U-Net-based diffusion models. **Secondly**, most prevailing quantization approaches heavily rely on post-training quantization (PTQ) (Li et al., 2023; He et al., 2024; Shang et al., 2023; Yang et al., 2023; Wang et al., 2023a), which leads to unacceptable performance degradation, particularly with extremely low bit width (e.g., 2-bit and 1-bit). For example, the 2-bit weight quantization results of Q-DiT (Chen et al., 2024) and Q-Diffusion (Li et al., 2023) are shown in Fig. 8. The extremely low-bit quantization of neural networks is important as it can significantly reduce the computation resources for deployment (Sze et al., 2017), especially for huge models. During our research, we find that there is no work considering the extremely low-bit quantization of DiT models.

To tackle these shortcomings, we propose to leverage the quantization-aware training (QAT) technique for the extremely low-bit quantization of large-scale DiT models. Low-bit QAT methods for large-scale models have been discussed in the LLM domain. Recent works observed that training large language models with extremely low-bit parameters (e.g., binary and ternary) from scratch can also lead to competitive performance (Wang et al., 2023b; Ma et al., 2024b) compared with their full-precision counterparts. This indicates that significant precision redundancy still exists in large-scale models, and it is feasible to conduct QAT for large-scale DiT models.

In this paper, we focus primarily on ternary weight networks (Li et al., 2016) and provide **TerDiT**, the first quantization scheme for extremely low-bit DiTs to our best knowledge. Our method achieves quantization-aware training (weight-only) and efficient deployment for ternary diffusion transformer models. Different from the naive quantization of linear layers in LLMs and CNNs (Ma et al., 2024b; Li et al., 2016), we find that the direct weight ternarization of the adaLN module (Perez et al., 2017) in DiT blocks (Peebles & Xie, 2023; Ma et al., 2024a) leads to large dimension-wise scale and shift values in the normalization layer compared with full-precision models (due to weight quantization, gradient approximation), which results in slower convergence speed and poor model performance. Consequently, we propose a variant of adaLN by applying an RMS Norm (Zhang & Sennrich, 2019) after the ternary linear layers of the adaLN module to mitigate this training issue effectively.

With this modification, we scale the ternary DiT model from 600M (size of DiT-XL/2 (Peebles & Xie, 2023)) to 4.2B (size of Large-DiT-4.2B (Gao et al., 2024)), with image resolution from 256×256 to 512×512. We further deploy the trained ternary DiT models with 2-bit CUDA kernels, resulting in over $10\times$ reduction in checkpoint size and about $8\times$ reduction in inference memory consumption, while achieving competitive (or even better) generation quality compared with full-precision models.

The contributions of our work are summarized as follows:

1. Inspired by the quantization-aware training scheme for low-bit LLMs, we study the QAT method for ternary DiT models and introduce DiT-specific techniques for achieving extremely low-bit quantization for DiT models.

2. We scale ternary DiT models from 600M to 4.2B parameters, with image resolution from 256×256 to 512×512, and further deploy the trained ternary DiT on GPUs based on 2-bit CUDA kernels, enabling the inference of a 4.2B DiT model with less than 2GB GPU memory (256×256 resolution).

3. Competitive evaluation results compared with full-precision models and baseline quantization methods on the ImageNet (Deng et al., 2009) benchmark (image generation) showcase the effectiveness of TerDiT.

TerDiT is the first attempt to explore the extremely low-bit quantization of DiT models. We focus on quantization-aware training and efficient deployment for large ternary DiT models, offering valuable insights for future research on deploying extremely low-bit DiT models.

## 2 RELATED WORKS

**Diffusion Models.** Diffusion models have gained significant attention in recent years due to their ability to generate high-quality images and their potential for various applications. The concept of diffusion models was first introduced by (Sohl-Dickstein et al., 2015), proposing a generative model that learns to reverse a diffusion process. This work laid the foundation for subsequent research in the field. (Ho et al., 2020) further extended the idea by introducing denoising diffusion probabilistic models (DDPMs), which have become a popular choice for image generation tasks. DDPMs have been applied to a wide range of domains, including unconditional image generation (Ho et al., 2020), image inpainting (Song et al., 2020), and image super-resolution (Saharia et al., 2021). Additionally, diffusion models have been used for text-to-image synthesis, as demonstrated by the DALL-E model (Ramesh et al., 2021) and the Imagen model (Saharia et al., 2022). These models showcase the ability of diffusion models to generate highly realistic and diverse images from textual descriptions. Furthermore, diffusion models have been extended to other modalities, such as audio synthesis (Chen et al., 2020) and video generation (Ho et al., 2022b), demonstrating their versatility and potential for multimodal applications.

**Quantization of Diffusion Models.** The quantization of diffusion models has been studied in recent years to improve the efficiency of diffusion models. Post-training quantization (PTQ) methods, such as those presented in (Li et al., 2023; He et al., 2024; Shang et al., 2023; Yang et al., 2023; Wang et al., 2023a; Chen et al., 2024; Deng et al., 2024), offer advantages in terms of quantization time and data usage. However, these methods often result in suboptimal performance when applied to low-bit settings. To address this issue, (He et al., 2023) proposes combining quantization-aware low-rank adapters (QALoRA) with PTQ methods, leading to improved evaluation results. As an alternative to PTQ, quantization-aware training (QAT) methods have been introduced specifically for low-bit diffusion model quantization (Wang et al., 2024; Zheng et al., 2024; Li et al., 2024a; Chang et al., 2023). Despite their effectiveness, these QAT methods are currently only limited to small-sized U-Net-based diffusion models, revealing a research gap in applying QAT to large-scale DiT models. Further exploration of QAT techniques for large DiT models with extremely low bit width could potentially unlock even greater efficiency gains and enable the effective deployment of diffusion models in resource-constrained environments.

**Ternary Weight Networks.** Ternary weight networks (Li et al., 2016) have emerged as a memory-efficient and computation-efficient network structure, offering the potential for significant reductions in inference memory usage. When supported by specialized hardware, ternary weight networks can also deliver substantial computational acceleration. Among quantization methods, ternary weight networks have garnered notable attention, with two primary approaches being explored: weight-only quantization and weight-activation quantization. In weight-only quantization, as discussed in (Zhu et al., 2016), solely the weights are quantized to ternary values. On the other hand, weight-activation quantization, as presented in (Alemdar et al., 2017; Wang et al., 2018), involves quantizing both the weights and activations to ternary values. Recent research has demonstrated the applicability of ternary weight networks to the training of large language models (Ma et al., 2024b), achieving results comparable to their full-precision counterparts. Building upon these advancements, our work introduces, for the first time, quantization-aware training and efficient deployment schemes specifically designed for ternary DiT models. By leveraging the benefits of ternary quantization in the context of DiT models, we aim to push the boundaries of efficiency and enable the deployment of powerful diffusion models in resource-constrained environments, opening up new possibilities for practical applications.

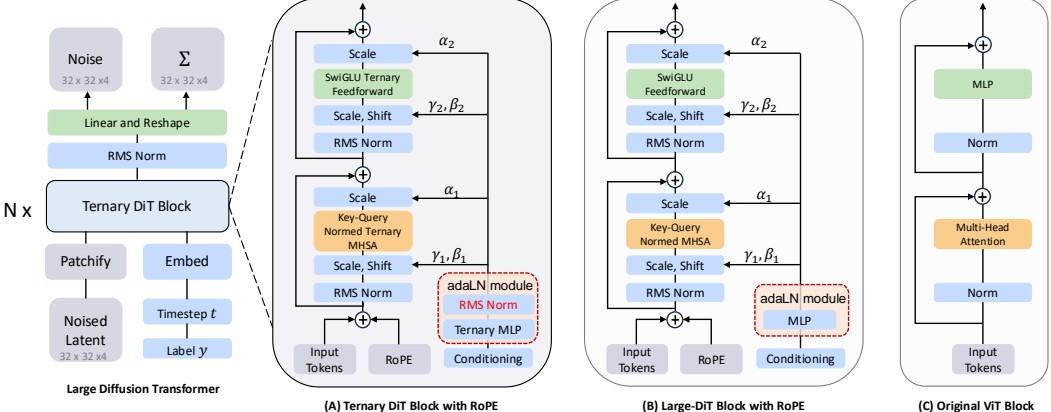

Figure 2: Model structure comparison between (A) Ternary DiT block, (B) Large-DiT block, and the (C) original ViT block. The Large-DiT (DiT) block adds an adaLN module to the original ViT block for condition injection. Ternary DiT block further adds an RMS Norm in the adaLN module for better ternarization-aware training.

## 3   TERDIT

In this section, we introduce TerDiT, a framework designed to conduct weight-only quantization-aware training and efficient deployment of large-scale ternary DiT models. We first give a brief review of diffusion transformer (DiT) models in Sec. 3.1. Then building upon the previous open-sourced Large-DiT (Gao et al., 2024), we illustrate the quantization function and quantization-aware training scheme in Sec. 3.2, conduct QAT-specific model structure improvement for better network training in Sec. 3.3, and introduce ternary deployment scheme in Sec. 3.4.

### 3.1   PRELIMINARY: DIFFUSION TRANSFORMER MODELS

**Diffusion Transformer.** Diffusion transformer (Peebles & Xie, 2023) (DiT) is an architecture that replaces the commonly used U-Net backbone in the diffusion models with a transformer that operates on latent patches. Similar to the Vision Transformer (ViT) architecture shown in Fig. 2 (C), DiT first pachifies the spatial inputs into a sequence of tokens, then the denoising process is carried out through a series of transformer blocks (Fig. 2 (B)). To deal with additional conditional information (e.g., noise timesteps $t$, class labels $l$, natural language inputs), DiT leverages adaptive normalization modules (Karras et al., 2019) (adaLN-Zero) to insert these extra conditional inputs to the transformer blocks. After the final transformer block, a standard linear decoder is applied to predict the final noise and covariance. The DiT models can be trained in the same way as U-Net-based diffusion models.

**AdaLN Module in DiT.** The main difference between DiT and traditional ViT is the need to inject conditional information for image generation. DiT employs a zero-initialized adaptive layer normalization (adaLN-Zero) module in each transformer block, as shown in the red part of Fig. 2 (B), which calculates the dimension-wise scale and shift values from the input condition $c$:

$$\text{adaLN}(c) = \text{MLP}(\text{SiLU}(c)). \tag{1}$$

AdaLN is an important component in the DiT model (Peebles & Xie, 2023) and has been proven more effective than cross-attention and in-context conditioning methods. Within the DiT architecture, the adaLN module integrates an MLP layer with a substantial number of parameters, constituting approximately 10% to 20% of the model's total parameters. Throughout the training of TerDiT, we observe that the direct weight-ternarization of this module yields undesirable training results (analyzed in Sec. 3.3).

### 3.2   MODEL QUANTIZATION

As illustrated in Sec. 1, there is an increasing popularity in understanding the scaling law of DiT models, which has been proven crucial for developing and optimizing LLMs. In recent explorations, Large-DiT (Gao et al., 2024) successfully scales up the model parameters from 600M to 7B by incorporating the methodologies of LLaMA (Touvron et al., 2023a;b) and DiT. The results demonstrate that parameter scaling can potentially enhance model performance and improve convergence speed for the

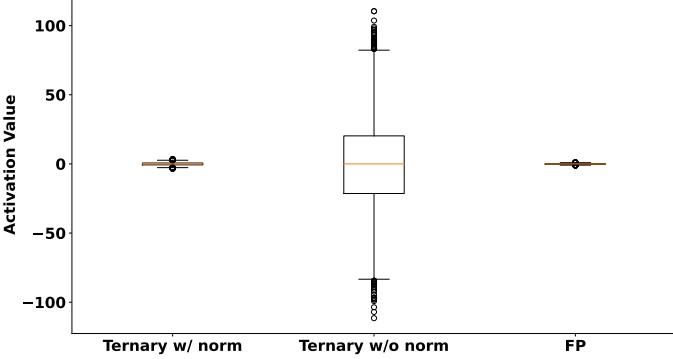

Figure 3: Activation value analysis. We compare activation values passing through a ternary weight linear layer with and without RMS Norm, using a full-precision linear layer as a reference. The ternary linear layer without RMS Norm results in extremely large activation values, introducing instability in neural network training. However, when the normalization layer is applied, the activation values are scaled to a reasonable range, similar to those observed in the full-precision layer.

label-conditioned ImageNet generation task. Motivated by this, we propose to further investigate the ternarization of DiT models, which can alleviate the challenges associated with deploying large-scale DiT models. In this subsection, we introduce the quantization function and quantization-aware training scheme.

**Quantization Function.** To construct a ternary weight DiT network, we replace all the linear layers in self-attention, feedforward, and MLP of the original Large-DiT blocks with ternary linear layers, obtaining a set of ternary DiT blocks (Fig. 2 (A)). For ternary linear layers, we adopt an *absmean* quantization function similar to BitNet b1.58 (Ma et al., 2024b). First, the weight matrix is normalized by dividing each element by the average absolute value of all the elements in the matrix. After normalization, each value in the weight matrix is rounded to the nearest integer and clamped into the set $\{-1, 0, +1\}$.

Referring to current popular quantization methods for LLMs (Frantar et al., 2022; Lin et al., 2023), we also multiply a learnable scaling parameter $\alpha$ to each ternary linear matrix after quantization, leading to the final value set as $\{-\alpha, 0, +\alpha\}$. The quantization function is formulated as:

$$\widetilde{W} = \alpha \cdot \text{RoundClip}\left(\frac{W}{\gamma + \epsilon}, -1, 1\right), \tag{2}$$

where $\epsilon$ is set to a small value (e.g., $10^{-6}$) to avoid division by 0, and

$$\text{RoundClip}(x, a, b) = \text{Clamp}(\text{round}(x), a, b) \text{ and } \gamma = \frac{1}{mn}\sum_{ij}|W_{ij}|. \tag{3}$$

TerDiT is a weight-only quantization scheme and we do not quantize the activations.

**Quantization-aware Training Scheme.** Based on the above-designed quantization function, we train a DiT model from scratch[1] utilizing the straight-through estimator (STE) (Bengio et al., 2013), allowing gradient propagation through the undifferentiable network components. We preserve the full-precision parameters of the network throughout the training process. For each training step, ternary weights are calculated from full-precision parameters by the ternary quantization function in the forward pass, and the gradients of ternary weights are directly applied to the full-precision parameters for parameter update in the backward pass.

However, we find the convergence speed is very slow. Even after many training iterations, the loss cannot be decreased to a reasonable range. We find that this issue may arise from the trait that ternary linear layers usually cause large activation values, and propose to tackle the problem with QAT-specific model structure improvement in the following subsection.

### 3.3 QAT-SPECIFIC MODEL STRUCTURE IMPROVEMENT

**Activation Analysis for Ternary Linear Layer.** In a ternary linear layer, all the parameters take one value from the set $\{-\alpha, 0, +\alpha\}$. The values passing through this layer would become large activation

---

[1]It is observed in (Ma et al., 2024b) that for ternary LLMs, the conversion or post-training quantization from trained LLMs does not help. So we also train ternary DiT models from scratch.

values, which might hamper the stable training of neural networks. We conduct a pilot study to qualitatively demonstrate the impact of ternary linear weights on activation values.

We randomly initialize a ternary linear layer with the input feature dimension set to 1024 and the output feature dimension to 9216 (corresponding to the linear layer of the adaLN module in Large-DiT). The weight parameters pass through the quantization function and receive a $512 \times 1024$ sized matrix input (filled with 1). The box plot of activation distribution is shown in the center part of Fig. 3. We also calculate the activation distribution after passing the matrix through a full-precision linear layer generated with the same random seed, shown on the right part of Fig. 3. As can be seen, the ternary linear layer leads to very large activation values compared with full-precision linear layers.

The large-activation problem brought about by the ternary linear weights can be alleviated by applying a layer norm to the output of the ternary linear layer. We add an RMS Norm (similar to LLaMA) after the ternary linear layer and obtain the activation distribution (as shown in the left part of Fig. 3). In this case, the activation values are scaled to a reasonable range after passing the normalization layer and lead to more stable training behavior. The observation also aligns with (Wang et al., 2023b), where a layer normalization function is applied before the activation quantization for each quantized linear layer.

**RMS Normalized AdaLN Module.** We analyze the DiT model for QAT-specific model structure improvement based on the above insights. In a standard ViT transformer block, the layer norm is applied for every self-attention layer and feedforward layer. This is also the case with the self-attention layers and feedforward layers in the DiT block, which can help to properly scale the range of activations. However, the DiT block differs from traditional transformer blocks due to the presence of the AdaLN module, as introduced in Sec. 3.1. Notably, there is no layer normalization applied to this module. In the context of full-precision training, the absence of layer normalization does not have a significant impact. However, for ternary DiT networks, its absence can result in large dimension-wise scale and shift values in the adaLN (normalization) module, posing bad influences on model training. To mitigate this issue, we introduce an RMS Norm after the MLP layer of the adaLN module in each ternary DiT block:

$$\text{adaLN\_norm}(c) = \text{RMS}(\text{MLP}(\text{SiLU}(c))), \tag{4}$$

and the final model structure of TerDiT is illustrated in Fig. 2 (A). This minor modification can result in faster convergence speed and lower training loss, leading to better quantitative and qualitative evaluation results. To better showcase the effect, the **actual** activation distribution with/without the RMS Norm after model training is analyzed in Sec. A.1.

**Remark: what are the differences between TerDiT and classic low-bit quantization approaches like BitNet (Wang et al., 2023b; Ma et al., 2024b)?**

In BitNet, a BitLinear layer is designed, which applies a layer norm to the input activation of the Linear layer. BitNet then replaces the Linear layers in LLaMA with BitLinear layers and removes the RMS Norm before the attention and SwiGLU layers. Similar approaches can be found in earlier works like Xnor-net (Rastegari et al., 2016), and Bi-real net (Liu et al., 2018).

Different from applying layer norm to all the BitLinear layers of the model, TerDiT focuses on **simple yet effective** modifications to the model structure by simply adding an RMS Norm within the adaLN module of each DiT block. With fewer norms added, our method leads to faster training speed and better evaluation scores. Comparative experiments are introduced in Sec. 4.2.

### 3.4 DEPLOYMENT SCHEME

After training the DiT model, we find that there are currently no effective open-source deployment solutions for ternary networks (Ma et al., 2024b). In this case, we deploy the trained networks with a 2-bit implementation. To be specific, we pack the ternary linear weights to int8 values (4 ternary numbers into one int8 number) with the `pack_2bit_u8()` function provided by (Badri & Shaji, 2023). During the inference process of the DiT model, we call the complementary `unpack_2bit_u8()` function on the fly to recover packed 2-bit numbers to floating-point values, then perform subsequent calculations. The addition of the unpacking operation will slow down the inference process, but we believe that with further research into model ternarization, more hardware support for inference speedup will become available.

| ImageNet 256×256 Benchmark | | | | | | |
|---|---|---|---|---|---|---|
| Models | Images (M) | FID ↓ | sFID ↓ | Inception Score ↑ | Precision ↑ | Recall ↑ |
| BigGAN-deep (Brock et al., 2018) | - | 6.95 | 7.36 | 171.40 | 0.87 | 0.28 |
| StyleGAN-XL (Sauer et al., 2022) | - | 2.30 | 4.02 | 265.12 | 0.78 | 0.53 |
| ADM (Dhariwal & Nichol, 2021) | 507 | 10.94 | 6.02 | 100.98 | 0.69 | 0.63 |
| ADM-U (Dhariwal & Nichol, 2021) | 507 | 7.49 | 5.13 | 127.49 | 0.72 | 0.63 |
| LDM-8 (Rombach et al., 2022) | 307 | 15.51 | - | 79.03 | 0.65 | 0.63 |
| LDM-4 (Rombach et al., 2022) | 213 | 10.56 | - | 103.49 | 0.71 | 0.62 |
| DiT-XL/2 (675M) (Peebles & Xie, 2023) | 1792 | 9.62 | 6.85 | 121.50 | 0.67 | 0.67 |
| **TerDiT-4.2B** | 604 | 9.66 | 6.75 | 117.54 | 0.66 | 0.68 |
| **Classifier-free Guidance** | | | | | | |
| ADM-G (Dhariwal & Nichol, 2021) | 507 | 4.59 | 5.25 | 186.70 | 0.82 | 0.52 |
| ADM-G, ADM-U (Dhariwal & Nichol, 2021) | 507 | 3.94 | 6.14 | 215.84 | 0.83 | 0.53 |
| LDM-8-G (Rombach et al., 2022) | 307 | 7.76 | - | 209.52 | 0.84 | 0.35 |
| LDM-4-G (Rombach et al., 2022) | 213 | 3.60 | - | 247.67 | 0.87 | 0.48 |
| DiT-XL/2-G (675M) (Peebles & Xie, 2023) | 1792 | 2.27 | 4.60 | 278.24 | 0.83 | 0.57 |
| Large-DiT-4.2B-G (Gao et al., 2024) | 435 | 2.10 | 4.52 | 304.36 | 0.82 | 0.60 |
| **TerDiT-600M-G** | 448 | 4.34 | 4.99 | 183.49 | 0.81 | 0.54 |
| **TerDiT-4.2B-G** | 604 | 2.42 | 4.62 | 263.91 | 0.82 | 0.59 |
| **ImageNet 512×512 Benchmark** | | | | | | |
| ADM (Dhariwal & Nichol, 2021) | 1385 | 23.24 | 10.19 | 58.06 | 0.73 | 0.60 |
| ADM-U (Dhariwal & Nichol, 2021) | 1385 | 9.96 | 5.62 | 121.78 | 0.75 | 0.64 |
| ADM-G (Dhariwal & Nichol, 2021) | 1385 | 7.72 | 6.57 | 172.71 | 0.87 | 0.42 |
| ADM-G, ADM-U (Dhariwal & Nichol, 2021) | 1385 | 3.85 | 5.86 | 221.72 | 0.84 | 0.53 |
| DiT-XL/2-G (Peebles & Xie, 2023) | 768 | 3.04 | 5.02 | 240.82 | 0.84 | 0.54 |
| Large-DiT-4.2B-G (Gao et al., 2024) | 472 | 2.52 | 5.01 | 303.70 | 0.82 | 0.57 |
| **TerDiT-4.2B-G** | 696 | 2.81 | 4.96 | 267.86 | 0.84 | 0.55 |

Table 1: Comparison between TerDiT and a series of full-precision diffusion models on the ImageNet $256 \times 256$ and $512 \times 512$ label-conditional generation task. For generation with classifier-free guidance, we use cfg=1.5. As can be seen, TerDiT achieves comparable results with full-precision models. On the ImageNet $512 \times 512$ task, TerDiT-4.2B-G outperforms DiT-XL/2-G in all aspects. It also surpasses Large-DiT-4.2B-G in terms of sFID and Precision.

## 4 EXPERIMENTS

In this section, a set of experiments are carried out to evaluate our proposed TerDiT. We compare TerDiT with full-precision diffusion models in Sec. 4.1, with other quantization methods (PTQ and QAT) in Sec. 4.2, carry out deployment efficiency comparison in Sec. 4.3, and illustrate the effectiveness of the RMS Normalized adaLN module (ablation study) in Sec. 4.4. Our DiT implementation is based on the open-sourced code of Large-DiT-ImageNet[2]. We conduct experiments on ternary DiT models with 600M (size of DiT-XL/2) and 4.2B[3] (size of Large-DiT-4.2B) parameters respectively.

### 4.1 COMPARISON WITH FULL-PRECISION MODELS

We provide quantitative and qualitative comparison results between TerDiT and representative full-precision models in this subsection.

**Experiment Setup.** Following the evaluation setting of the original DiT paper (Peebles & Xie, 2023), we train 600M and 4.2B ternary DiT models on the ImageNet dataset. We start from training the 256×256 image resolution model (600M and 4.2B) and continue to train the 512×512 resolution model (4.2B) based on the 256×256 checkpoint. We compare TerDiT with a series of full-precision diffusion models, report FID (Heusel et al., 2017), sFID (Salimans et al., 2016), Inception Score, Precision, and Recall (50k generated images) following (Dhariwal & Nichol, 2021). We also provide the total number of images (million) during the training stage as in (Gao et al., 2024) to offer further insights into the convergence speed of different generative models.

**Training Details. 256×256 Resolution**: We train the 600M TerDiT model on 8 A100-80G GPUs for 1750k steps with batchsize set to 256, and the 4.2B model on 16 A100-80G GPUs for 1180k

---

[2]https://github.com/Alpha-VLLM/LLaMA2-Accessory/tree/main/Large-DiT-ImageNet

[3]The provided 3B model in the Large-DiT-ImageNet repository actually has 4.2B parameters.

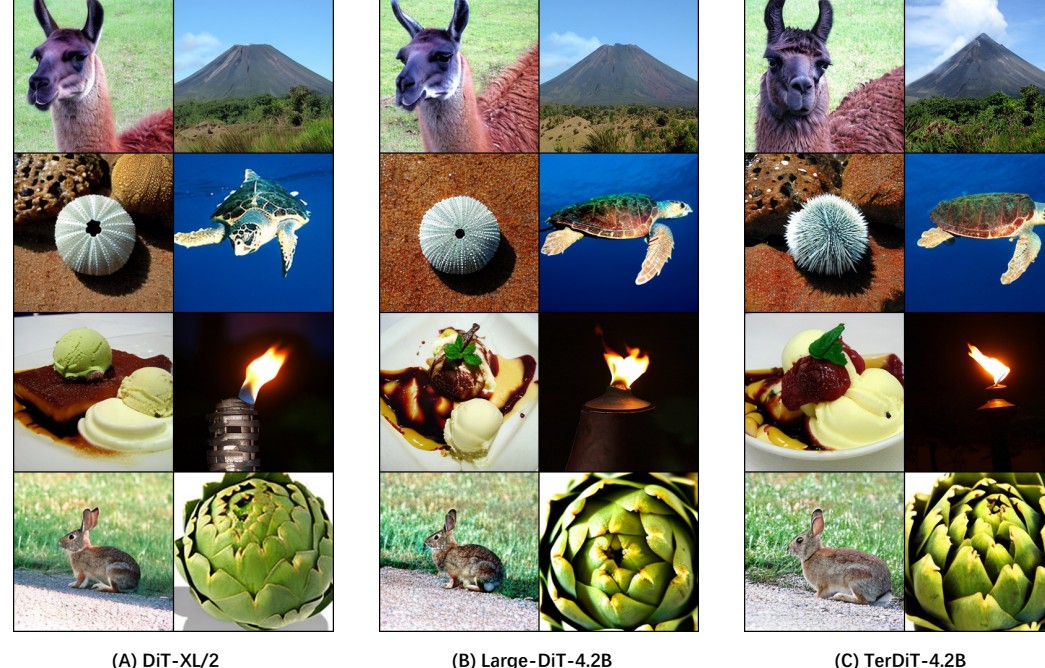

(A) DiT-XL/2        (B) Large-DiT-4.2B        (C) TerDiT-4.2B

Figure 4: Qualitative results analysis (256×256). We compare images generated by DiT-XL/2 (A), Large-DiT-4.2B (B), and TerDiT-4.2B (C) with class labels [355, 980, 328, 33, 928, 862, 330, 944] and cfg=4. TerDiT-4.2B generates images of the same quality as two full-precision DiT models.

steps with batchsize set to 512. We set the initial learning rate as 5e-4[4]. After 1550k steps of training for the 600M and 550k steps for the 4.2B model, we reduce the learning rate back to 1e-4 for more fine-grained parameter updates (ablation study on this learning rate reduction is provided in Sec. A.3).

**512×512 Resolution**: Considering 512×512 resolution is the more commonly used resolution for generation models, we study the generalization ability of TerDiT to 512×512 resolution. Starting from the 4.2B trained 256×256 ternary model (trained with 604M images), we continue to train it on ImageNet with 512×512 resolution and only 92M images. The learning rate is set to 1e-4.

**Quantitative Results Analysis.** The evaluation results are listed in Tab. 1. TerDiT is a QAT scheme for DiT models, so among all the full-precision models, we pay special attention to DiT-XL/2 (675M) and Large-DiT-4.2B. (Large-DiT-4.2B and TerDiT-4.2B-G share the exact same architecture.)

In the 256×256 case, without classifier-free guidance, TerDiT-4.2B achieves very similar testing results with DiT-XL/2 (with much fewer training images). With classifier-free guidance (cfg=1.5), TerDiT-4.2B-G outperforms LDM-G while bringing a very slight performance degradation compared with two full-precision DiT-structured models. Besides, TerDiT-4.2B-G achieves better evaluation results than TerDiT-600M-G, implying that models with more parameters can incur smaller performance degradation after quantization. In the more commonly used 512×512 setting, TerDiT-4.2B-G outperforms DiT-XL/2-G in all aspects of the ImageNet 512×512 task. It also surpasses Large-DiT-4.2B-G in terms of sFID and Precision. This result demonstrates the generative capabilities of TerDiT compared with full-precision models with the same architecture and parameter numbers.

**Qualitative Results Analysis.** To visually demonstrate the effectiveness of TerDiT, we also show some qualitative comparison results in Fig. 4 (256×256), concerning TerDiT-4.2B, DiT-XL/2 and Large-DiT-4.2B. In terms of visual perception, there is no significant difference between the images generated by TerDiT and those generated by the full-precision models. 512×512 and more 256×256 qualitative results are shown in Sec. A.5.

## 4.2 COMPARISION WITH OTHER EXTREMELY LOW-BIT QUANTIZATION BASELINES

The topic of our paper is to explore an algorithm for the ternary (1.58-bit) quantization of extremely low-bit DiT models. In this section, we make comparison between TerDiT and PTQ/QAT baselines.

---

[4]The default learning rate of DiT is 1e-4. For TerDiT, we increase the initial learning rate to 5e-4 following Wang et al. (2023b) that a larger learning rate is needed for the faster convergence of low-bit QAT.

|  | Resolution | Checkpoint Size | Max Memory Allocated | Inference Time | FID |
|---|---|---|---|---|---|
| DiT-XL/2-G | 256 | 2.6GB | 3128.42MB | 15s | 2.27 |
| DiT-XL/2-G | 512 | 2.6GB | 3728.61MB | 80s | 3.04 |
| Large-DiT-4.2B-G | 256 | 16GB | 17027.29MB | 83s | 2.10 |
| Large-DiT-4.2B-G | 512 | 16GB | 18614.29MB | 365s | 2.52 |
| TerDiT-600M-G | 256 | 168M | 762.30MB | 20s | 4.34 |
| TerDiT-4.2B-G | 256 | 1.1GB | 1919.67MB | 97s | 2.42 |
| TerDiT-4.2B-G | 512 | 1.1GB | 3506.67MB | 376s | 2.81 |

Table 2: Deployment efficiency comparison with cfg=1.5. TerDiT achieves a significant reduction in model size and memory usage while maintaining competitive evaluation results. Although the inference time of TerDiT is slightly slower due to the unpack operations, the inference time is expected to be significantly reduced with hardware support specifically designed for ternary models.

**PTQ Baselines at Extremely Low-bit Width.** We choose PTQ algorithms Q-DiT (Chen et al., 2024) for DiT models and Q-Diffusion (Li et al., 2023) for U-Net-based models and perform 2-bit weight quantization. We find they both fail to generate images, as detailed in Sec. A.2 and Fig. 8.

**QAT Baselines at Extremely Low-bit Width.** We adapt BitNet b1.58 (Wang et al., 2023b; Ma et al., 2024b) for the DiT model (256×256, 600M parameters). We also use Efficientdm (He et al., 2023) for a more comprehensive analysis. For BitNet, we replace the linear layers in DiT (which correspond to TerDiT) with BitLinear and remove the norms before attention and SwiGLU layers. We then train BitNet using the same process as TerDiT and measure the FiD score ($\downarrow$), which yields 6.60 for BitNet and 4.34 for TerDiT. Moreover, due to the additional norms, the training speed of BitNet drops to $0.9\times$ that of TerDiT, while inference latency increases to $1.15\times$. This demonstrates the efficiency of our proposed TerDiT. For Efficientdm, we apply 2-bit quantization and train the quantized model, but it fails to generate normal images, as detailed in Sec. A.2 and Fig. 9.

## 4.3 DEPLOYMENT EFFICIENCY COMPARISON

The improvement in deployment efficiency is the motivation of our proposed TerDiT scheme. In this subsection, we provide a comparison between TerDiT-600M/4.2B, DiT-XL/2, and Large-DiT-4.2B to discuss the actual deployment efficiency TerDiT can bring about. Tab. 2 shows the checkpoint sizes of the four DiT models. We also record the memory usage and inference time of the total diffusion sample loop (step=250) on a single A100-80G GPU.

From the table, we can see that TerDiT greatly reduces checkpoint size and memory usage. The checkpoint size and memory usage of the 4.2B ternary DiT model are significantly smaller than those of Large-DiT-4.2B, even smaller than DiT-XL/2. This brings significant advantages to deploying the model on end devices (e.g., mobile phones, FPGA). However, due to the absence of open-source software deployment frameworks and efficient hardware support (Ma et al., 2024b) for ternary DiTs, we observe slower inference speeds compared to their full-precision counterparts. Despite this, we anticipate that the computational benefits of ternary-weight networks will become more apparent with future advancements in software and hardware co-design.

## 4.4 DISCUSSION ON THE RMS NORMALIZED ADALN MODULE

The main modification of TerDiT to the structure of the DiT model is the addition of an RMS Norm after the MLP in the adaLN module. In this part, we compare with the baseline ternary model to demonstrate the influence of RMS Norm on both the training process and the training outcomes.

**Experiment Setup.** We train ternary DiT models with 600M and 4.2B parameters on the ImageNet (Deng et al., 2009) dataset in 256×256 resolution. For each parameter size, we train two models, one with RMS Norm in the adaLN module and one without (our baseline). We record the loss curves during training and measure the FID-50k score (cfg=1.5) every 100k training steps.

**Training Details.** For a fair comparison, we train all the ternary DiT models on 8 A100-80G GPUs with batchsize set to 256. The learning rate is set to 5e-4 throughout the training process.

**Results Analysis.** The training loss[5] and evaluation scores are shown in Fig. 5 and Fig. 6 respectively. As can be seen, training with the RMS Normalized adaLN module will lead to faster convergence

---

[5]The actual training process of diffusion models is not as 'smooth' as one might assume. To better illustrate the training dynamics, we employ exponential smoothing (with a smoothing factor of 0.995) during visualization.

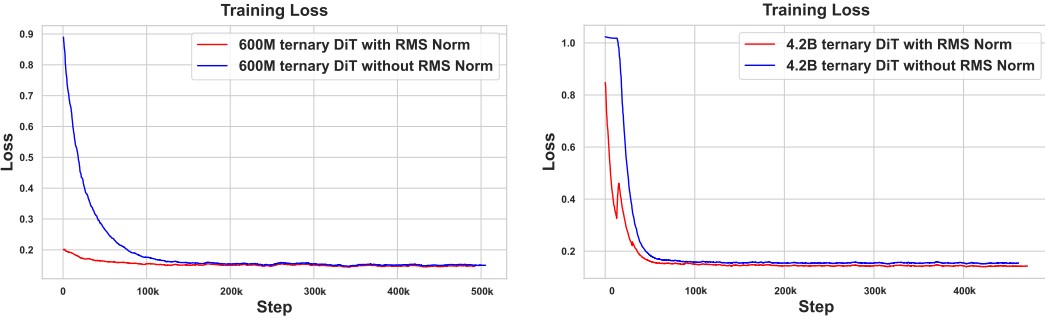

Figure 5: Training loss comparison of ternary DiT models with/without RMS Norm in the adaLN module. We show the loss curves training both 600M (left) and 4.2B (right) DiT models. Adding the RMS Norm will lead to faster convergence speed and lower training loss.

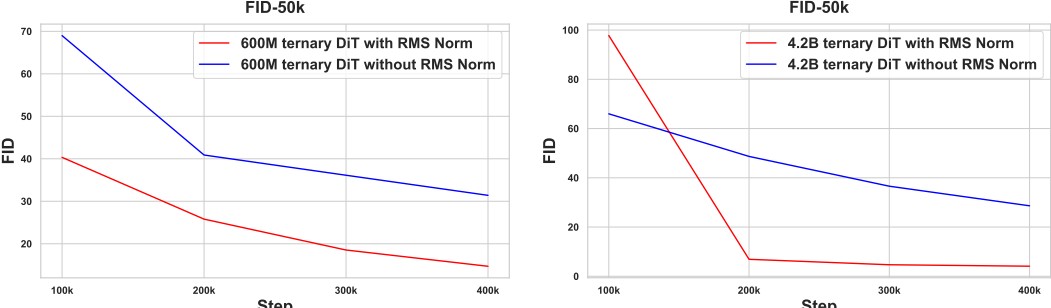

Figure 6: FID-50k score comparison on class-conditional ImageNet 256×256 generation task (cfg=1.5) with/without RMS Norm for both 600M and 4.2B ternary DiT models (100k steps to 400k steps). Training with RMS Norm will lead to lower FID scores.

speed and lower FID scores. Another observation is that models with more parameters tend to achieve faster and better training compared to models with fewer parameters. This also to some extent reflects the scaling law of the ternary DiT model. Qualitative comparison results are shown in Sec. A.4.

## 5 DISCUSSIONS AND FUTURE WORKS

In this paper, we propose quantization-aware training (QAT) and efficient deployment methods for large-scale ternary DiT models. Competitive evaluation results on the ImageNet generation task (256×256 and 512×512) compared with full-precision models and baseline quantization methods demonstrate the feasibility of training a large ternary DiT from scratch while achieving promising generation results. To our best knowledge, this work is also the first study concerning the extremely low-bit quantization of DiT models. In this section, we give more explanations and discussions.

**Firstly**, training ternary DiT models is less stable and more time-consuming than training full-precision networks. Although we discuss methods to enhance training stability by adding norms, it still requires more time than training full-precision networks, leading to increased carbon dioxide emissions during model training in a broader context. We plan to further explore hardware-software co-development to increase training speed in the future.

**Secondly**, in our paper, we do not make a complete comparison between ternary quantization and INT8/FP16 quantization, as INT8 and FP16 do not fall under the category of extremely low-bit quantization, which is outside the comparison scope of our paper. On the one hand, hardware and software support for INT8 and FP16 is mature, with many hardware chips and software libraries readily available (Micikevicius et al., 2017; Frantar et al., 2022; Dettmers et al., 2022; Lin et al., 2023). In contrast, research on extremely low-bit quantization for LLMs or Stable Diffusion remains in its early stages. On the other hand, while FP16 or INT8 can achieve strong performance, they reduce memory usage by only up to 75%. In theory, ternary quantization can reduce memory usage by up to 16×. Therefore, it is undeniable that ternary quantization holds greater potential compared to FP16 or INT8. In fact, hardware and software accelerations for ternary (binary) CNNs have already been implemented on FPGA (Zhu et al., 2022; Zhao et al., 2017; Rutishauser et al., 2024). We will continue to explore the hardware acceleration implementation of TerDiT in future work.

We hope our work can serve as a pipeline for the extremely low-bit quantization for diffusion transformers, and inspire the community to engage in this area, fostering broader advancements (e.g., software and hardware co-design) in this field of study.

REPRODUCIBILITY STATEMENT

In this paper, we utilize the widely recognized ImageNet dataset for training. The training and inference code for TerDiT is provided in the supplementary material. Detailed implementation information, including hyperparameters, is outlined in Sec. 4. Furthermore, we conduct a comprehensive set of comparisons and ablation studies to assess the effectiveness of TerDiT. We believe this will be valuable for future research endeavors.

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

# A    APPENDIX / SUPPLEMENTAL MATERIAL

## A.1    ACTIVATION DISTRIBUTION AFTER TRAINING

Here we conduct an analysis of the activation distribution during inference after model training, as a supplementary for the experiment and explanations in Sec. 3.3.

Following Sec. 4.4, we train the TerDiT-4.2B model (256×256) with/without RMS Norm in the adaLN module for 50k steps. We then analyze the activation distribution of the 'scale_mlp' output in the adaLN module during inference, specifically at the first sampling step within the second ternary DiT block. For comparison, we also calculate the activation distribution of the original full-precision Large-DiT-4.2B model at the same layer. As shown in Fig. 7, training with RMS Norm can help limit the range of the activation values.

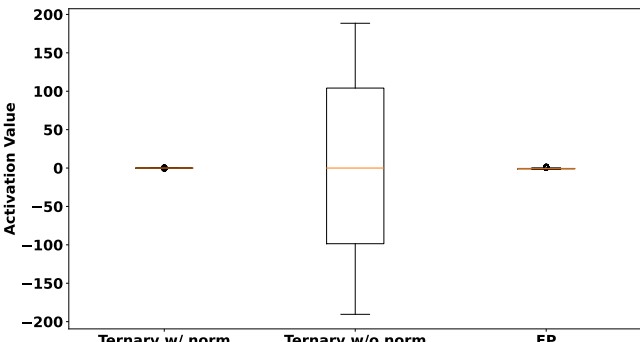

Figure 7: Activation value analysis. We train the TerDiT-4.2B model with/without RMS Norm in the adaLN module for 50k steps and show the activation distribution of the 'scale_mlp' output (one output of the adaLN module) in the second ternary DiT block during inference (at the first sampling step). The activation distribution of the original full-precision model is also provided.

## A.2 COMPARISON WITH OTHER PTQ AND QAT BASELINES

**PTQ Baselines at Extremely Low-bit Width:** We show the 2-bit weight quantization results of Q-DiT (Chen et al., 2024) (for DiT models) and Q-Diffusion (Li et al., 2023) (for U-Net-based models) at 256×256 resolution. These two methods both fail at the extremely low-bit setting.

(a) 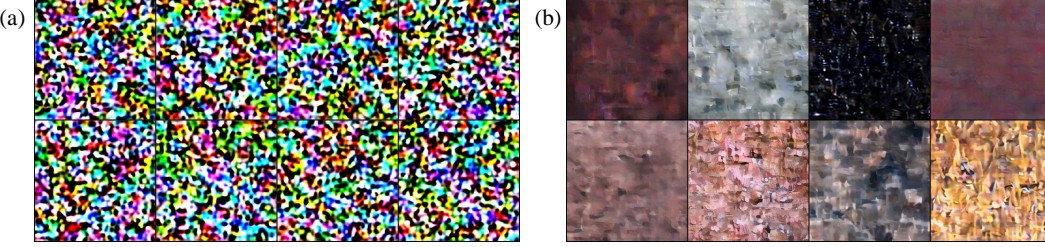 (b)

Figure 8: 2-bit Q-DiT quantization results (a) and 2-bit Q-Diffusion quantization results (b).

**QAT Baselines at Extremely Low-bit Width:** We show the 2-bit weight quantization results of EfficientDM (He et al., 2023) at 256×256 resolution. We find it just fails to generate reasonable images.

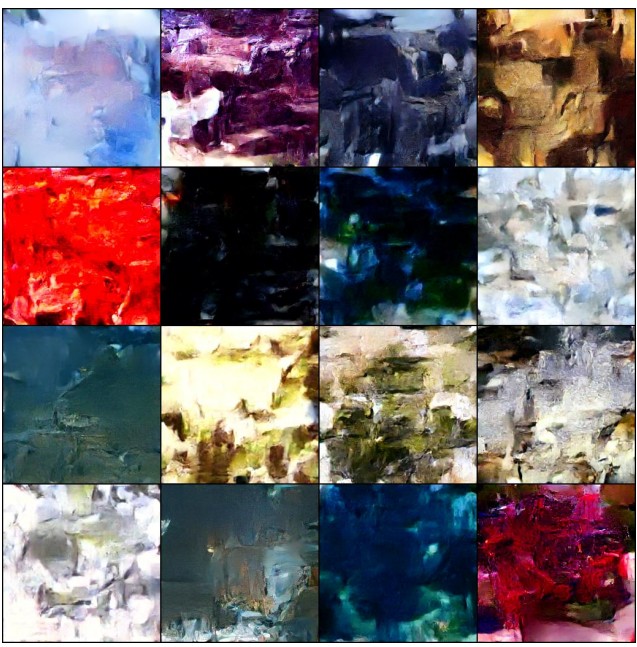

Figure 9: 2-bit weight quantization results of Efficientdm. Efficientdm fails to generate images.

## A.3 EFFECTIVENESS OF LEARNING RATE REDUCTION

In Sec. 4.1, we adopt a learning rate reduction after training for certain steps for more fine-grained parameter updates. Here we provide an ablation study on this learning rate reduction.

We choose the TerDiT-600M model (256×256) for convenience. Following the setting of Sec. 4.1, we train a TerDiT-600M model with RMS Normalized adaLN module and 5e-4 learning rate for 1550k steps. We then continue training this model with 1e-4/5e-4 for another 200k steps.

**Quantitative Results.** We evaluate FID, sFID, Inception Score, Precision, and Recall of these two models. As can be seen in Tab. 3, the reduction in learning rate will lead to better evaluation results.

| ImageNet 256×256 Benchmark, Classifier-free Guidance | | | | | | |
|---|---|---|---|---|---|---|
| Models | LR | FID $\downarrow$ | sFID $\downarrow$ | Inception Score $\uparrow$ | Precision $\uparrow$ | Recall $\uparrow$ |
| **TerDiT-600M-G** (cfg=1.50) | 1e-4 | 4.34 | 4.99 | 183.49 | 0.81 | 0.54 |
| **TerDiT-600M-G** (cfg=1.50) | 5e-4 | 6.38 | 5.00 | 147.79 | 0.76 | 0.54 |

Table 3: Effectiveness of learning rate reduction. Training with a reduced learning rate after 1550k steps for the TerDiT-600M model will lead to better evaluation results.

**Qualitative Results.** We also provide qualitative results comparison on TerDiT-600M (256×256) with/without learning rate reduction in Fig. 10. For reference, the image generated by TerDiT-4.2B (fully trained, 256×256) on the provided class label is shown in Fig. 11. The quality comparison highlights the importance of learning rate reduction in the later stages of training.

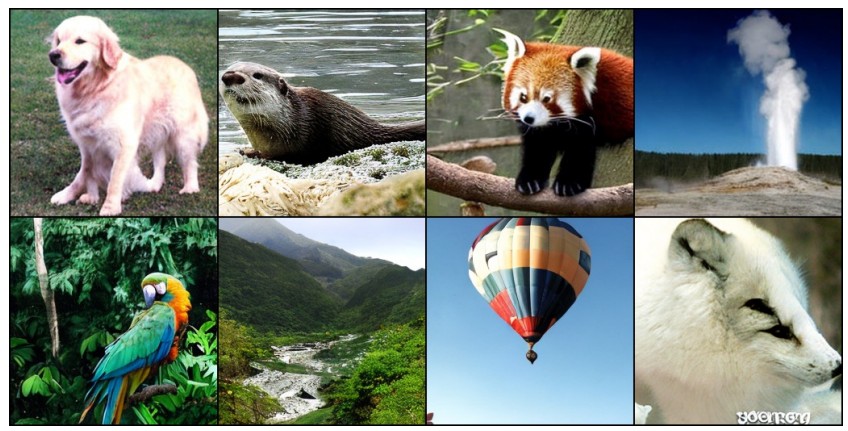

**TerDiT-600M without LR reduction**

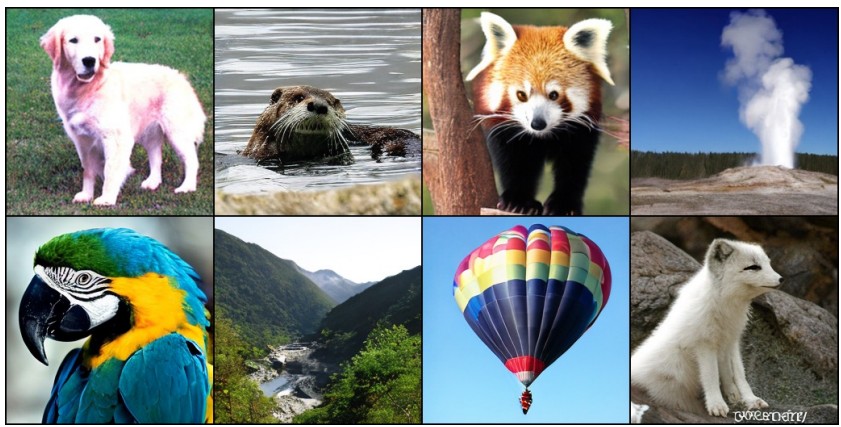

**TerDiT-600M with LR reduction**

Figure 10: TerDiT-600M, class label [207, 360, 387, 974, 88, 979, 417, 279], cfg=4.

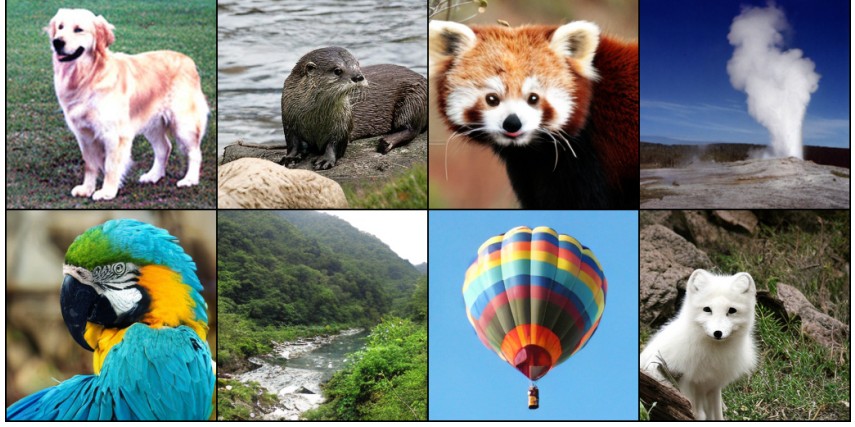

Figure 11: TerDiT-4.2B, class label [207, 360, 387, 974, 88, 979, 417, 279], cfg=4.

## A.4 QUALITATIVE RESULTS ANALYSIS FOR RMS NORMALIZED ADALN MODULE

In Sec. 4.4 we provide quantitative comparison results for models trained with/without RMS Norm in the adaLN module (256×256). Here we provide more qualitative results with models trained for 400k steps, together with the fully trained models.

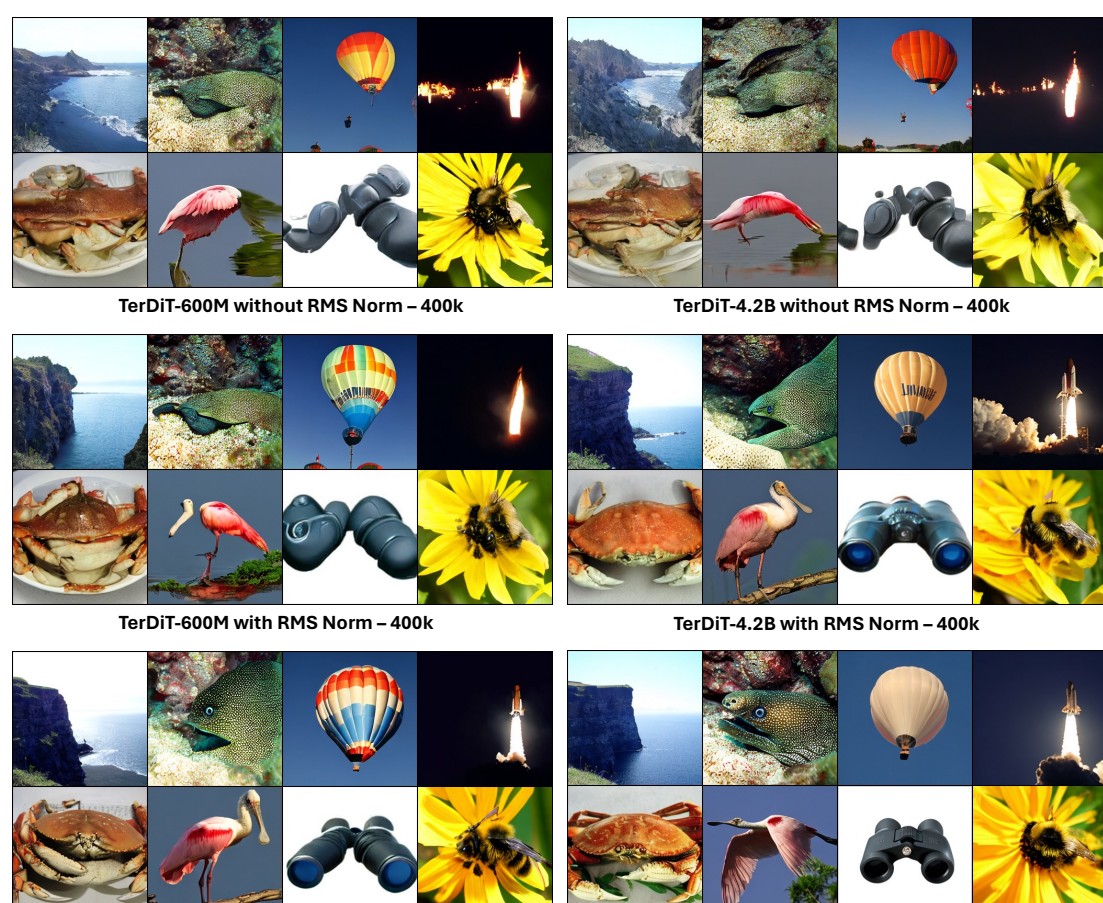

Figure 12: Qualitative results comparison for 600M and 4.2B ternary DiT models with/without RMS Norm in the adaLN module. We choose class labels [972, 390, 417, 812, 118, 129, 447, 309] with cfg=4. Training the 4.2B model with the RMS Normalized adaLN module will lead to better qualitative generation results.

We compare TerDiT-600M and TerDiT-4.2B models trained with/without RMS Norm in the adaLN module. We sample images from the models trained for 400k steps in Sec. 4.4 and also show the results of the fully trained models. The comparisons are demonstrated in Fig. 12.

We can come to two conclusions:

1. For both the 600M and 4.2B TerDiT model, training with the RMS Normalized adaLN module will lead to better qualitative results.

2. Models with more parameters show more learning ability and can achieve better training results compared to models with fewer parameters.

These observations are consistent with the quantitative results provided in Sec. 4.4.

### A.5 MORE QUALITATIVE RESULTS

**256×256**:

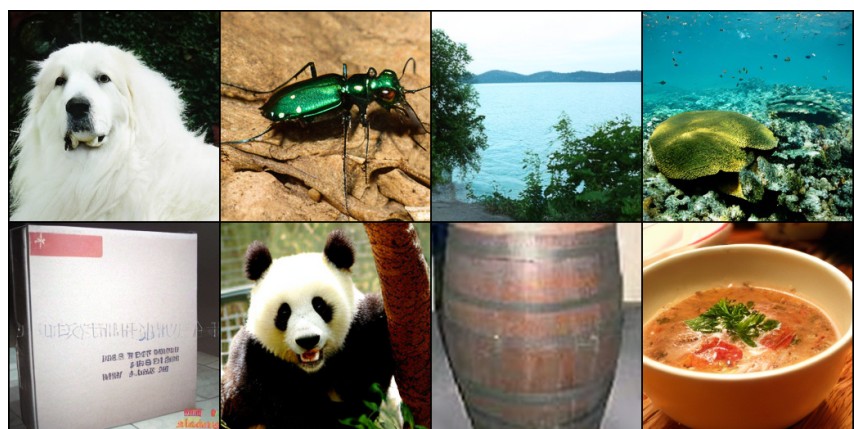

Figure 13: TerDiT-4.2B, class label [257, 300, 975, 973, 478, 388, 427, 809], cfg=4.

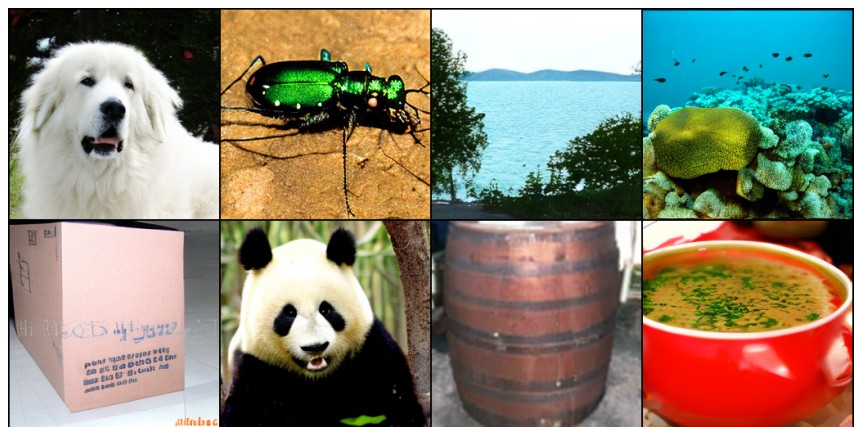

Figure 14: TerDiT-600M, class label [257, 300, 975, 973, 478, 388, 427, 809], cfg=4.

**512×512**:

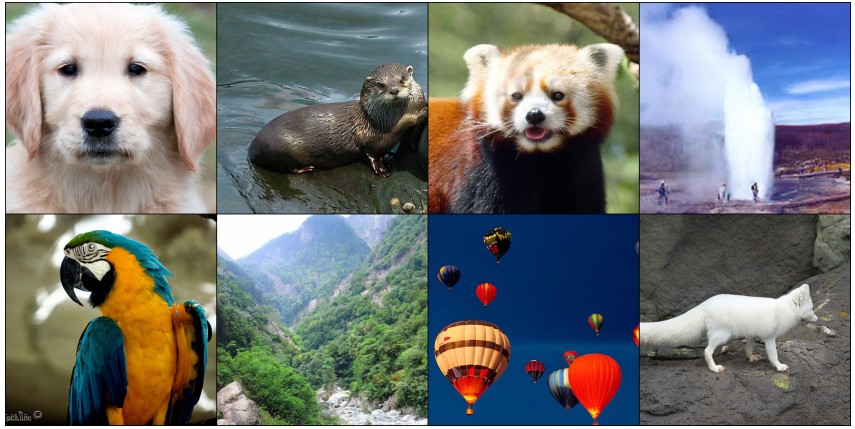

Figure 15: TerDiT-4.2B, class label [207, 360, 387, 974, 88, 979, 417, 279], cfg=4.

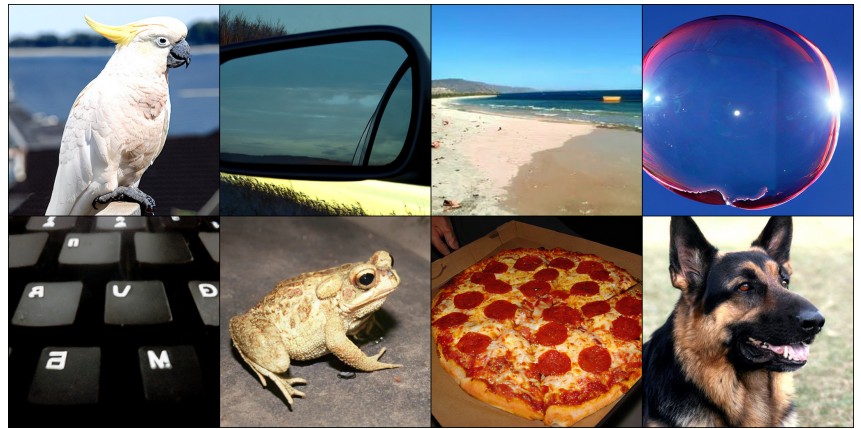

Figure 16: TerDiT-4.2B, class label [89, 475, 978, 971, 508, 32, 963, 235], cfg=4.

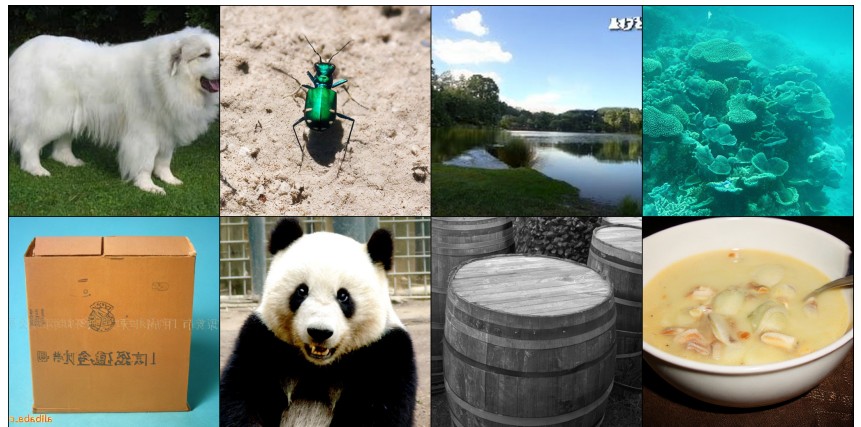

Figure 17: TerDiT-4.2B, class label [257, 300, 975, 973, 478, 388, 427, 809], cfg=4.

