# OpenReview forum: "TerDiT: Ternary Diffusion Models with Transformers"
_ICLR.cc/2025/Conference — ICLR 2025 Conference Withdrawn Submission_

### Official Review · Reviewer_9vbQ · 2024-10-27

**Soundness:** 2
**Presentation:** 2
**Contribution:** 1
**Rating:** 3
**Confidence:** 5

**Summary:**

This work proposes a Quantization Aware Training (QAT) paradigm to obtain the proposed ternaryDiT models. By adding an additional RMSNorm layer after the AdaLN of the DiT model, the proposed method can effectively overcome the issue of large activation values. The authors evaluate the performance of the proposed method on image generation tasks with different resolutions and across various model sizes(600M and 4.2B).

**Strengths:**

- The proposed method achieves the ternarization of DiT model through the QAT technique, enhancing the efficiency of the DiT model.

- The additional RMSNorm after AdaLN to address the issue of large activation values is simple and efficient.

**Weaknesses:**

- The technical contribution of this paper appears to be merely the introduction of an extra RMSNorm. The CUDA kernel deployment is from previous work, and the model architecture follows the previous Large-DiT. Therefore, I rise concerns about the technical contribution of this paper.

- The models architecture evaluated in this work are somewhat limited. For example, previous work on diffusion model quantization included various diffusion architectures, such as LDM, StableDiffusion, etc. However, this paper, which focus on the DiT quantization, only considers the Large-DiT architecture. Considering the diversity of diffusion transformers, such as VDT[1], evaluating only one architecture weakens the contribution of this paper. Additionally, this also raises the question of whether the large activation values problems is a common phenomenon for diffusion transformers?

- In Table 2, the authors provide a comparison of the efficiency, and it can be seen that TerDiT achieves lower checkpoint size and GPU memory usage. However, the proposed method does not significantly increase the inference time compared to the full-precision model. I have noted the authors' explanation for this, but in practical applications, inference speed is a very important factor we care about. The proposed method does not truly improve inference speed, which weakens the significance of the quantization of DiT.

- Why adding RMSNorm can alleviate large activation values? The paper only provides empirical results; a more in-depth theoretical or visualization analysis is necessary for understanding the working mechanism.

-  In Table 1, why are the performance of TerDiT-600M-G is only given under the CFG settings?

> [1] VDT: General-purpose Video Diffusion Transformers via Mask Modeling. ICLR24

**Questions:**

- The proposed method focuses solely on weight quantization. Since this paper uses more computationally expensive QAT, considering activation quantization would further benefit this work.

-  It is well-known that DiT is famous for its role as a video generation backbones; how effective of this method in accelerating video generation models?

- Why use RMSNorm instead of other normalization mehotds?

- Overall, although this work claims to be the first to study extremely low-bit quantization of DiT, in my view, this work is not in-depth and solid. It merely adds an extra RMSNorm as its technical contribution. Therefore, I am tend to give a score of reject.

---

### Official Review · Reviewer_Noo3 · 2024-10-29

**Soundness:** 3
**Presentation:** 4
**Contribution:** 3
**Rating:** 5
**Confidence:** 5

**Summary:**

This paper extends the concept of 1.58-bit LLMs, which were introduced earlier this year [1], to Diffusion Transformers [2]. Specifically, 1.58-bits refers to ternary weights - each weight is either -1, 0 or 1. Specific contributions include the addition of RMSNorm in several components of the DiT including the Adaptive LayerNorm (AdaLN), usage of 2-bit CUDA kernels displaying model checkpoint size savings, and experimental results on ImageNet.

**Strengths:**

- Competent QAT method for Diffusion Transformers.
- QAT jumps to very low bit, ternary quantization.
- Experimental results, when presented, are convincing.
- The paper is well-written, easy to understand, and competently presented.
- Visual results per Fig. 4 and otherwise are impressive.

**Weaknesses:**

- The evaluation on ImageNet is incomplete. Specifically, the authors consider two resolutions (256 and 512, corresponding to the original DiT) and two model sizes (600M param XL/2 and a sized-up 4.2B parameter version), but their application of the two models is inconsistent across experiments, e.g., the 600M model is only used for 256x256 for CFG in Table 1.
- Table 1 results are decently convincing but not great, e.g., L337 TerDiT-600M loses substantially to the original DiT-XL/2 (almost double FID, almost half Inception Score) and at 512 resolution TerDiT-4.2-G is not much better than the much smaller DiT-XL/2 despite being much larger.
- For all the trashing of PTQ said in L072, there is inadequate comparison to PTQ methods. Yes, PTQ methods like Q-DiT [3] and Q-Diffusion [4] fail at 2-bit precision (really anything below 4-bits), this is not a novel finding, especially for ViT-like architectures [5]. The trade-off between QAT and PTQ has always been that the former allows lower bit precision at the cost of much higher computational resources.
- No activation quantization. The 1.58-bit LLM paper [1] tried for 8-bit activations in addition to Ternary weights. Moreover, even if PTQ methods are limited to ~4 bit weights, these methods generally attempt activation quantization to at least 8-bits. This makes a comparison crucial as it could even the odds in terms of inference cost (e.g., BitOPs, latency, etc.) compared to this QAT method.
- There is at least one factual inaccuracy in the writing at lines 203-205: "AdaLN is an important component in the DiT model (Peebles & Xie, 2023) and has been proven more effective than cross-attention and in-context conditioning methods." this is absolutely not true and must be corrected. While the original DiT model removes Cross-Attention for AdaLN, it is very limited and restricted to ImageNet generation. T2I DiTs like the PixArt [6, 7] and Hunyuan [8] re-introduce cross-attention while SD3/3.5 and Flux [9] concat the text and image sequences together which is similar.
- L260 - 263: I find this line of reasoning to not be borne out by the experimental results and ablations, e.g., Figs 5/6. In Fig 5 the loss of the DiT without RMS Norm is higher initially but quickly drops to be comparable. Fig 6 shows RMS Norm achieves faster FID reduction, however, FID itself is not entirely a trustworthy metric on its own [10].

**Questions:**

No questions, but more nitpicks and comments:
- L192: 'pachifies'
- Fig 2 - I do not see why this figure needs to be as large and detailed as it is just for some simple changes.
- Fig 3 too much whitepsace.
- Eq. 4 "adaln_norm" spells out as adaptive layer normalization normalization... change.

References:

[1] Ma, Shuming, et al. "The era of 1-bit llms: All large language models are in 1.58 bits." arXiv preprint arXiv:2402.17764 (2024).

[2] Peebles, William, and Saining Xie. "Scalable diffusion models with transformers." Proceedings of the IEEE/CVF International Conference on Computer Vision. 2023.

[3] Chen, Lei, et al. "Q-dit: Accurate post-training quantization for diffusion transformers." arXiv preprint arXiv:2406.17343 (2024).

[4] Li, Xiuyu, et al. "Q-diffusion: Quantizing diffusion models." Proceedings of the IEEE/CVF International Conference on Computer Vision. 2023.

[5] Frumkin, Natalia, Dibakar Gope, and Diana Marculescu. "Jumping through local minima: Quantization in the loss landscape of vision transformers." Proceedings of the IEEE/CVF International Conference on Computer Vision. 2023.

[6] Chen, Junsong, et al. "Pixart-$\alpha $: Fast training of diffusion transformer for photorealistic text-to-image synthesis." arXiv preprint arXiv:2310.00426 (2023).

[7] Chen, Junsong, et al. "Pixart-\sigma: Weak-to-strong training of diffusion transformer for 4k text-to-image generation." arXiv preprint arXiv:2403.04692 (2024).

[8] Li, Zhimin, et al. "Hunyuan-DiT: A Powerful Multi-Resolution Diffusion Transformer with Fine-Grained Chinese Understanding." arXiv preprint arXiv:2405.08748 (2024).

[9] Esser, Patrick, et al. "Scaling rectified flow transformers for high-resolution image synthesis." Forty-first International Conference on Machine Learning. 2024.

[10] Podell, Dustin, et al. "Sdxl: Improving latent diffusion models for high-resolution image synthesis." arXiv preprint arXiv:2307.01952 (2023).

---

### Official Review · Reviewer_uq4v · 2024-11-01

**Soundness:** 2
**Presentation:** 3
**Contribution:** 2
**Rating:** 3
**Confidence:** 3

**Summary:**

The manuscript presents the first ternary-weight DiT, applying a ternary quantizer to DiT and incorporating RMS Norm into the adaLN module, achieving high-accuracy image generation through QAT training.

**Strengths:**

1.	This is the first attempt to ternarize DiT, achieving generative results that can even rival those of full-precision models.

2.	Compared to BitNet b1.58, the manuscript selectively applies RMS Norm based on the structural characteristics of DiT.

3.	Comparative results across multiple models confirm the effectiveness of TerDiT.

**Weaknesses:**

1. The paper lacks a clear motivation for applying a ternary quantizer to DiT. Given effective precedents for binary DM and 2-bit DM, I believe evidence is needed to demonstrate that a ternary DM can significantly outperform binary DM in accuracy or achieve substantial efficiency gains over 2-bit DM in practical use.

2. The issue of increased activations does not necessarily seem to be an inherent problem of ternarization. For example, better initialization of α in Equation 2 might straightforwardly address the phenomena shown in Figure 3.

3. The comparison with EfficientDM appears unfair. EfficientDM’s QAT training is based on LoRA, which has a natural disadvantage in precision. For a more compelling comparison with QAT methods, alternatives like LSQ and Q-DM [1] (including noise-estimating mimicking) would be more appropriate.

4. Table 1 compares only parameter counts but does not list the actual size, memory, or OPs.

5. The phrase "significantly reduced" in the caption of Table 2 lacks factual basis, as practical acceleration gains from weight-only binary quantizers are generally limited. The extent to which a carefully designed ternary quantizer could deliver such gains remains uncertain.

6. There is a lack of comparative data for using RMSNorm across the full model. In the Remark section at the end of Section 3.3, the manuscript states, "Comparative experiments are introduced in Sec.4.2," but I could not find any supporting experiments anywhere in the manuscript.

[1] Li, et al. Q-dm: An efficient low-bit quantized diffusion model.

**Questions:**

1. Why choose a ternary quantizer for DiT rather than a 2-bit quantizer?

Would a 2-bit quantizer for DiT potentially yield better results than the ternary approach?

As the manuscript’s deployment relies on 2-bit operators, why insist on using 3-bit?

2. How is α in Equation 2 initialized?

3. Would using RMS Norm for the entire model achieve better results? If so, does this mean the strategy improvement in this manuscript is limited compared to BitNet b1.58?

---

### Official Review · Reviewer_eHqz · 2024-11-03

**Soundness:** 3
**Presentation:** 3
**Contribution:** 2
**Rating:** 5
**Confidence:** 3

**Summary:**

The paper proposes TerDiT, a quantization aware training process that could be leverage to perform memory efficient training of DIT based architectures. The paper also introduces an efficient deployment scheme for ternary DITs. For this the authors introduce Ternary DiT blocks that replaces the conventional DiT blocks during training. The overall idea is to perform 1 bit bit quantization at different layers of the diffusion model during the training process.  The authors also noted that a naive replacement of the attention and feed forward layers in DiT leads to high activations which worsens performance. Hence they introduce an RMS normalization to alleviate the effect. The proposed changes bring in a reduction of maximum memory allocated  by a factor of 6 and checkpoint size by a factor of 14 while causing minor reduction in FID scores.

**Strengths:**

1. The paper is the first to utilize Ternary blocks in Diffusion transformers for quantization aware training.
2. The paper identifies a major problem in post training quantization of DiT and illustrates the drastic drop in performance that happens when about 1-2 bits of post training based quantization is used.
3. The paper identifies a major problem that happens in low bit quantization aware training using Ternary blocks, which is the high activation values and proposes a simple fix based on LLAMA to solve this problem. The authors also show that the with the utilization of RMSNorm, the performance in terms of FID drastically improves for TerDiT.

**Weaknesses:**

1. The method lacks in novelty and the main contribution of the paper is in utilizing the ideas in [1] for replacing the attention and feedforward blocks and [2] for introducing RMSNorm for stable training process.
2. The results in Figure 6 are surprising since the training losses with and without RMSNorm converge to similar values but the FID performance shows a drastic difference. Could the authors comment further on this.
3. In Figure 6, could the authors also please provide the results with classifier free guidance if it was already not performed.
4. Ln 303: The authors may have missed out to point the different between the work and [3]. Could the authors please highlight the major differences with [1].

[1] Shuming Ma, Hongyu Wang, Lingxiao Ma, Lei Wang, Wenhui Wang, Shaohan Huang, Li Dong, Ruiping Wang, Jilong Xue, and Furu Wei. The era of 1-bit llms: All large language models are in 1.58 bits. arXiv preprint arXiv:2402.17764, 2024b

[2] Touvron, H., Lavril, T., Izacard, G., Martinet, X., Lachaux, M.A., Lacroix, T., Rozière, B., Goyal, N., Hambro, E., Azhar, F. and Rodriguez, A., 2023. Llama: Open and efficient foundation language models. arXiv preprint arXiv:2302.13971.

**Questions:**

1. In addition there are some small grammatical errors like
  a. pachifies in Ln 192

---

### Note · Authors · 2024-11-13

I have read and agree with the venue's withdrawal policy on behalf of myself and my co-authors.